# Towards Multimodal-augmented Pre-trained Language Models via Self-balanced Expectation-Maximization Iteration

## ABSTRACT

Pre-trained language models (PLMs) that rely solely on textual corpus may present limitations in multimodal semantics comprehension. Existing studies attempt to alleviate this issue by incorporating additional modal information through image retrieval or generation. However, these methods: (1) inevitably encounter modality gaps and noise; (2) treat all modalities indiscriminately; and (3) ignore visual or acoustic semantics of key entities. To tackle these challenges, we propose a novel principled iterative framework for multimodal-augmented PLMs termed MASE, which achieves efficient and balanced injection of multimodal semantics under the proposed Expectation Maximization (EM) based iterative algorithm. Initially, MASE utilizes multimodal proxies instead of explicit data to enhance PLMs, which avoids noise and modality gaps. In E-step, MASE adopts a novel information-driven self-balanced strategy to estimate allocation weights. Furthermore, MASE employs heterogeneous graph attention to capture entity-level fine-grained semantics on the proposed multimodal-semantic scene graph. In M-step, MASE injects global multimodal knowledge into PLMs through a cross-modal contrastive loss. Experimental results show that MASE consistently outperforms competitive baselines on multiple tasks across various architectures. More impressively, MASE is compatible with existing efficient parameter fine-tuning methods, such as prompt learning.

## CCS CONCEPTS

• **Theory of computation** → *Theory and algorithms for application domains*; *Machine learning theory*.

## KEYWORDS

Multimodal-augmented, Pre-trained Language Models, Self-balancing optimization, Entity-level enhancements

## 1 INTRODUCTION

Leveraging self-supervised techniques, pre-trained language models (PLMs) such as BERT [9], T5 [30], GPT [3], have achieved strong progress towards comprehending human language. The paradigm of fine-tuning PLMs has achieved tremendous success in various downstream NLP tasks [3, 26, 28]. However, most existing PLMs rely predominantly on contextual learning which only takes the textual context as self-supervision [25, 38]. These language learners

Permission to make digital or hard copies of all or part of this work for personal or classroom use is granted without fee provided that copies are not made or distributed for profit or commercial advantage and that copies bear this notice and the full citation on the first page. Copyrights for components of this work owned by others than the author(s) must be honored. Abstracting with credit is permitted. To copy otherwise, or republish, to post on servers or to redistribute to lists, requires prior specific permission and/or a fee. Request permissions from permissions@acm.org.

*ACM MM, 2024, Melbourne, Australia*

© 2024 Copyright held by the owner/author(s). Publication rights licensed to ACM.
ACM ISBN 978-x-xxxx-xxxx-x/YY/MM
https://doi.org/10.1145/nnnnnnn.nnnnnnn

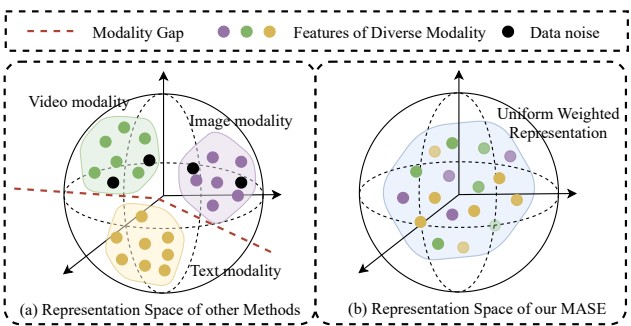

**Figure 1: Comparison between our MASE and other methods. Most existing methods treat various modal information indiscriminately and inevitably encounter modality gaps and noise. Our MASE can achieve a uniform representation space while achieving self-balancing weighting of various modal information.**

overlook the potential of learning from multimodal corpora, e.g., image, audio and video [2] and cannot well capture multimodal-grounded semantics, e.g., visual commonsense [17]. Moreover, prior studies have proved that these PLMs exhibit human biases [12] and lack general multimodal knowledge [50], e.g., color and shape.

To overcome the above challenges, existing methods mainly enhance the visual comprehension ability of PLMs by retrieving or synthesizing real images. These methods employ images to enhance PLMs during the pre-training [39, 43] or the fine-tuning stage [15, 25], or leverage images to bolster the zero-shot capabilities of PLMs [48]. Despite the notable progress achieved by these studies, we identify several key issues: **(1) Modality Gap and Noise**: As shown in Figure 1, they necessitate labor-intensive image retrieval or generation and are prone to introducing irrelevant noise, which can degrade the performance. Furthermore, it is widely recognized that a large semantic gap [21] exists between different modalities, which will further hinder modal aggregation. **(2) Modality Imbalance**: They focus exclusively on image information while overlooking audio and video modalities, and handle all modal information indiscriminately. Recent work [7, 15] reveals that the relevance of various modal types varies across specific NLP tasks, and not all information from additional modalities proves beneficial. Consequently, the indiscriminate treatment of all modal information is suboptimal. **(3) Ignoring Entity-level Multimodal Semantics**: As shown in Figure 2, from the perspective of the scene graph, we discover that key entities are highly correlated with additional modal information, such as 'dog' corresponding to visual features, and 'called' corresponding to audio features. Previous work overlooked fine-grained visual or acoustic semantics correlated with these key objects, which are crucial for improving the multimodal understanding of entities in PLMs.

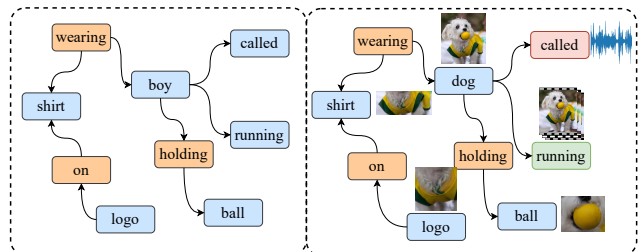

**Figure 2: We parse the corpus into a semantic scene graph (left) and realize the absence of entity-level multimodal semantics. To capture multimodal-semantic fusion representations, our MASE innovatively constructs an entity-level multimodal-semantic scene graph (right).**

In this paper, we propose a novel principled EM-based iteration framework for self-balancing multimodal-augmented PLMs (**MASE**). As illustrated in Figure 1, MASE tackles the above challenges through proposed strategies in the following three aspects:

(1) **MASE utilizes multimodal proxies as a bridge for information transmission and employs an EM-based iteration for optimization.** MASE initially leverages textual instances to obtain implicit modal proxies through multimodal contrastive pre-training models (MC-PTMs) (e.g., CLIP [29], AudioCLIP [16] and CLIP-ViP [47], etc.). These proxies are intrinsically aligned with real-world data and enable the circumvention of noise and modality gaps. Subsequently, MASE conceptualizes the balance and efficient injection of multimodal information from proxies as a probabilistic problem, and adopts the EM algorithm [8] for optimization.

(2) **MASE utilizes a novel information-driven strategy for allocation estimation.** Specifically, MASE modals information allocation weights as latent variables under our iteration framework, adopts the proposed mutual information (MI) based method to estimate the contribution of various modalities to the targets in the E-step. By our MI-based estimation approach, we can calculate a specified posterior probability of assigned weights to explore the intrinsic structure of multimodal information, achieve multimodal balance in the optimization process.

(3) **MASE achieves entity-level multimodal information enhancement via the proposed multimodal-semantic scene graph (MSSG).** The multimodal representation of entities (including objects and attributes) and relations can reveal a large amount of world-commonsense information and rich visual-spatial semantics, which helps to enhance the multimodal understanding ability of PLMs. To fully exploit fine-grained cross-modal features at the entity level, We innovatively concatenate multimodal features with entity nodes in the vanilla scene graph to construct MSSG, and propose employing heterogeneous graph attention to efficiently extract enhanced representation upon MSSG in E-step.

Following the balance and enhancement in the E-step, we utilize cross-modal contrastive learning to achieve global optimization of MASE in the M-step. Extensive experiments conducted on thirteen datasets across five tasks demonstrate the effectiveness and universality of MASE. In addition, MASE can be combined with existing efficient parameter fine-tuning strategies, such as prompt learning. In general, our contributions are three-fold:

(**Theory**) We propose a principled iterative framework termed MASE that utilizes the EM-based algorithm to inject multimodal semantics into PLMs balance and efficiently. To our knowledge, it is the first theoretical framework for multimodal-augmented PLMs.

(**Methodology**) We innovatively propose an MI-based information allocation estimation method and an entity-level multimodal aggregation method via the proposed MSSG to achieve modal semantic balance and enhancement, respectively.

(**Experiments**) Extensive experiments on five tasks (including NLP, visual reasoning, and visual question answering) demonstrate the effectiveness and generality of MASE. More impressively, our method is compatible with different architectures and existing efficient parameter fine-tuning methods.

## 2 RELATED WORK

**Pre-trained Language Models**. Recent advancements in large-scale PLMs have been achieved through self-supervised learning techniques applied to extensive text corpora [3, 9, 30]. These PLMs exhibit robust generalization capabilities, and fine-tuning them significantly enhances the performance of various downstream tasks, including NLU, question answering, and text generation. It has been observed that language models trained exclusively on textual data may not adequately grasp multimodal contexts and world knowledge, such as visual commonsense [12, 48, 50]. Moreover, recent studies indicate that merely expanding the textual corpus does not overcome these limitations [14, 50]. In this article, we introduce a universal probability framework designed to enrich the multimodal semantic integration within PLMs.

**Multimodal Contrastive Pre-trained Models.** MC-PTMs are developed through training on a substantial corpus of modality-pairing samples, enabling the mapping of various modalities into a unified representation space. CLIP [29] undergoes pre-training with a vast collection of image-text pairs, effectively achieving semantic alignment between images and text. AudioCLIP [16] and CLIP-ViP [47] expand CLIP-based approach to the audio modality and video domain, respectively. These works inspire us to obtain semantics aligned with real modal data through textual instances to enhance multimodal understanding of PLMs.

**Multimodal-Augmented Language Models.** Most studies focus on incorporating visual modality information into PLMs through retrieval or image generation methods. Tan and Bansal [39], Wang et al. [43] incorporate visual knowledge in the pre-training phase of PLMs. Guo et al. [15], Lu et al. [25] enhance PLMs by integrating visual information during the fine-tuning process. Furthermore, Yang et al. [48] employs image generation or retrieval methods to encode visual representations, aiming to improve zero-shot NLU. As a comparison, MASE is the first theoretical framework for MA-PLMs, which can simultaneously achieve multimodal semantic injection and modal information balance.

## 3 METHODS

In this section, we first introduce the task setting and the multimodal proxy based baseline models for MA-PTMs in Section 3.1, and then describe our proposed probabilistic model in Section 3.2 for infusing multimodal knowledge into PLMs during fine-tuning.

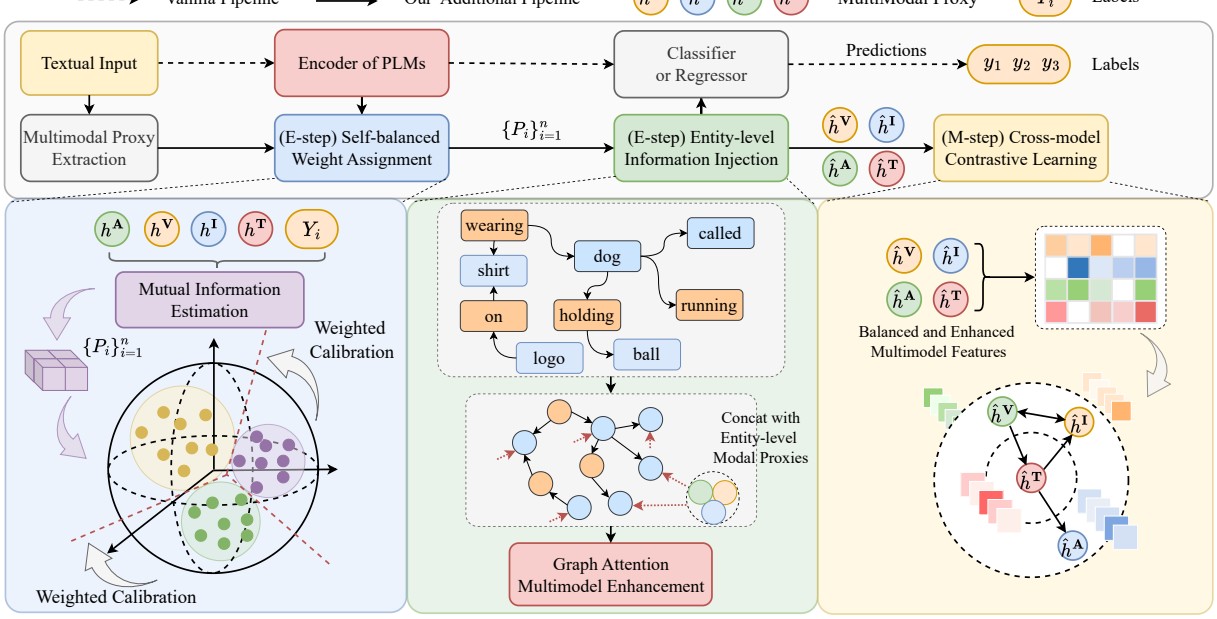

**Figure 3: The illustration of the proposed framework which is iteratively optimized by EM-based algorithms. In E-step, MASE uses our MI-driven estimation strategy to estimate modal allocation weights and achieves entity-level multimodal semantics injection via MSSG. In M-step, MASE achieves global optimization via cross-entropy and cross-modal contrastive losses.**

Subsequently, we describe the EM-based iterative optimization algorithm and proposed information-driven self-balanced allocation estimation strategy in Section 3.3 and Section 3.4, respectively. Finally, we summarize our total training algorithm in Algorithm 1.

### 3.1 Task Settings and Baseline Models

**Task Settings.** In general NLP scenarios, PLMs are adapted through fine-tuning on a specific dataset $\mathcal{D} = \{x_i, y_i\}_{i=1}^n$, aimed at executing tasks like classification or regression, where, $x_i$ and $y_i$ is the $i$-th textual instance and its label, respectively, with $n$ denoting the total number of training instances. Our study focuses on devising an efficient framework for integrating multi-modal information into PLMs throughout the fine-tuning process. The set of modalities engaged in this paper is denoted as $\mathcal{M} = \{\mathbf{T}, \mathbf{I}, \mathbf{A}, \mathbf{V}\}$, representing text, image, audio, and video modalities, correspondingly.

**Proxy-based Baseline Models for MA-PLMs.** As the multimodal proxy based multimodal-augmented PLM approach has not been studied in NLP tasks, we extend the vision-enhanced PLM approach, namely VAWI [15], to a multimodal setting and combine with MC-PTMs to achieve multimodal proxy based multimodal information extraction. MC-PTMs perform contrastive learning across a large number of cross-modal data pairs, and their text features are highly aligned with other modal features. Thus, the text semantics of MC-PTMs contain modal-specific representation encodings and can serve as implicit proxies. In practice, we employ the CLIP-based text encoder of MC-PTMs to facilitate implicit multimodal information as $h_i^m = \text{MC-PTMs}^m(x_i)$, where, $m$ denotes the represents a specific modality, i.e., $m \in \mathcal{M} \backslash \mathbf{T}$. In this work, we utilize CLIP [29], AudioCLIP [16] and CLIP-ViP [47] to obtain hidden state

encodings of image, audio, and video modality proxies, respectively. Subsequently, we concatenate different modal semantics (i.e., $h_i^m, \forall m \in \mathcal{M}$) and feed them into a classifier or regression to obtain predictions. We further provide theoretical analysis in supplements about our multimodal proxy is an excellent multimodal information transmission bridge.

### 3.2 Probabilistic Modeling for our MASE

Our MASE, as shown in Figure 3, adaptively assigns soft weights to different modal semantics $h_i^m$, where the lower weights resort to weakening the negative impact of irrelevant modalities. Specifically, MASE prioritizes the weight of information assignment as a latent variable and then constructs a general probabilistic model for various PLMs as follows:

**Latent Variables for Information Assignment.** We consider the weight of information assignment as a latent variable $z_i$ which represents the degree of contribution of different modalities to specific predictions. The latent variable $z_i = \{z_i^m\}$ as a measure of information allocation will directly act on the multimodal semantics $h_i^m, \forall m \in \mathcal{M}$ to obtain the optimal representation $h_i^*$:

$$h_i^* = \sum_{m}^{\mathcal{M}} z_i^m h_i^m. \tag{1}$$

**Probabilistic Model with Latent Variables.** Based on the multimodal representations $\mathbf{h}_i = \{h_i^m\}$ and latent variables $z_i$, we can optimize the model by maximizing the following log-likelihood:

$$\arg\max_{\Theta} \ell(\Theta, z) = \arg\max_{\Theta} \sum_{i=1} \log \sum_{z_i} p(y_i, z_i \mid \mathbf{h}_i; \Theta). \tag{2}$$

Optimizing the log-likelihood as presented in Eq. 2 poses a challenge due to the necessity of summing across logarithmic configurations of $z_i$. Fortunately, this optimization problem can be elegantly solved by maximizing a lower bound related to $z_i$. Specifically, we define a new discrete distribution $P_i(z_i)$ satisfying as:

$$\sum_{z_i} P_i(z_i) = 1, 0 \leqslant P_i(z_i) \leqslant 1, \tag{3}$$

where $\Theta$ denotes the learnable parameters of the model. Based on this, we can use Jensen's inequality [19] to derive the lower bound $\mathcal{F}(\Theta, \{P_i\}_{i=1}^{n})$ of the log-likelihood of Eq. 2 as:

$$\begin{aligned}
\ell(\Theta, z) &= \sum_{i=1}^{n} \log \sum_{z_i \in \{0,1\}} P_i(z_i) \frac{p(\hat{y}_i, z_i \mid \mathbf{h}_i; \Theta)}{P_i(z_i)} \\
&\geqslant \sum_{i=1}^{n} \sum_{z_i \in \{0,1\}} P_i(z_i) \log \frac{p(\hat{y}_i, z_i \mid \mathbf{h}_i; \Theta)}{P_i(z_i)} \\
&= \mathcal{F}(\Theta, \{P_i\}_{i=1}^{n}),
\end{aligned} \tag{4}$$

where we abbreviate $P_i(z_i)$ as $P_i$.

Eq. 4 inspires us to optimize the latent variables $z_i$ and the model parameters $\Theta$ by optimizing the lower bound $\mathcal{F}(\Theta, \{P_i\}_{i=1}^{n})$. This approach addresses the challenges associated with optimizing the likelihood function $\ell(\Theta, z)$. In this work, we propose a strategy based on the Expectation-Maximization (EM) algorithm to optimize this lower bound.

## 3.3 Overall EM-based Iteration for Optimization

MASE iteratively optimizes $\Theta$ and $\{P_i\}_{i=1}^{n}$ through E-step and M-step to achieve the balance of information allocation and effective injection of multimodal information, respectively.

**E-step for Self-balanced Information Assignment.** In E-step, we keep the current parameters $\Theta^{(t-1)}$ fixed and and maximize the lower bound $\mathcal{F}$ to estimate the allocation weights of different modal information as follows:

$$P_i(z_i)^{(t)} := p\left(z_i \mid \mathbf{h}_i, y_i; \Theta^{(t-1)}\right). \tag{5}$$

We will specifically discuss our information-driven allocation method for estimating $p\left(z_i \mid \mathbf{h}_i, y_i; \Theta^{(t-1)}\right)$ in Section 3.4.

**E-step for Entity-level Information Injection.** After obtaining the balanced modal features $h_i^m, \forall m \in \mathcal{M}$, we employ the proposed MSSG and heterogeneous graph attention networks to obtain enhanced multimodal features to $\hat{h}_i^m$, as detailed in Section 3.5.

**M-step for Multimodal Information Injection.** In this step, we hold $P_i(z_i)^{(t)}$ and update $\Theta$ to maximize the lower bound for injecting multimodal information as:

$$\Theta^{(t)} = \arg\max_{\Theta} \sum_{i=1}^{n} \sum_{z_i} P_i(z_i)^{(t)} \log p\left(y_i, z_i \mid \hat{\mathbf{h}}_i; \Theta\right). \tag{6}$$

In practice, MASE achieves the optimization goal of M-step in Eq. 6 by performing cross-entropy loss and cross-modal contrastive loss on the aggregated multimodal features $\hat{h}_i^*$ via Eq. 1. The cross-entropy loss is used to represent prediction error as:

$$\mathcal{L}_{ce}^{i} = \text{CE}\left(y_i; p\left(y_i \mid \hat{h}_i^*, z_i\right)\right), \tag{7}$$

where $\text{CE}(\cdot)$ denotes the cross-entropy function. And the cross-modal contrastive loss is used to optimize the representation between modalities, which can be represented as:

$$\mathcal{L}_{cc}^{i} = \sum_{m \in \mathcal{M}} \sum_{k \in \mathcal{M} \setminus \{m\}} \frac{\exp\left(\hat{h}_i^m \cdot \hat{h}_i^k / \tau\right)}{\sum_{j=1}^{n} \exp\left(\hat{h}_i^m \cdot \hat{h}_j^k / \tau\right)}, \tag{8}$$

where $\tau$ is the temperature coefficient.

Specifically, we minimize the following total loss function in M-step to inject balanced multimodal knowledge into PLMs:

$$\mathcal{L}_M = \frac{1}{n} \sum_{i} (\mathcal{L}_{ce}^{i} + \alpha \mathcal{L}_{cc}^{i}), \tag{9}$$

where $\alpha$ denotes a trade-off coefficient.

The optimization loss $\mathcal{L}_M$ incorporates strategies for allocating multimodal information via $z_i$ to achieve an adaptive balance and inject knowledge into PLMs effectively during the M-step phase. During training, we initialize $z_i$ to a uniform distribution and then iterate on E-step and M-step until convergence.

## 3.4 MI-driven Self-balanced Weight Allocation

In Section 3.3, it remains a crucial question of how to estimate the distribution of multimodal information allocation latent variables $p(z_i \mid \mathbf{h}_i, y_i; \Theta)$ in the E-step. We propose a novel MI-driven multimodal knowledge allocation strategy for dynamically estimating the distribution of latent variables. Specifically, we calculate the mutual information between different modal semantics and task labels in E-step as a measure of modalities' importance. This strategy is based on intuition: the greater the contribution of modal knowledge to task targets, the greater the mutual information. We mathematically define the mutual information between specific modal knowledge $h^m$ and labels $y$ as:

$$\mathcal{I}(h^m, y) = \mathcal{H}_{\Theta}(y) - \mathcal{H}_{\Theta}(y \mid h^m), \tag{10}$$

where $\mathcal{H}_{\Theta}(\cdot)$ denotes information entropy under the model $\Theta$. In practice, MASE adopts a Monte Carlo estimator to estimate the mutual information of random variables in a batch. The mutual information $\mathcal{I}(h^m, y)$ has a lower bound and an upper bound, i.e., $0 \leq \mathcal{I}(h^m, y) \leq \mathcal{H}_{\Theta}(y)$. $\mathcal{H}_{\Theta}(y)$ is a fixed value representing the information entropy of the task dataset labels. Based on the above analysis, we can utilize normalized mutual information to establish an information allocation distribution model $p\left(z_i \mid \mathbf{h}_i, y_i; \Theta^{(t)}\right)$ for $z_i = \{z_i^m\}$ as:

$$p(z_i^m \mid \mathbf{h}_i, y_i; \Theta) = \frac{\mathcal{I}(h^m, y)}{\sum_m^{\mathcal{M}} \mathcal{I}(h^m, y)}, \tag{11}$$

where $m \in \mathcal{M}$ denotes different modalities. We achieve the dynamic balance of multimodal information in the iterative optimization by using our MI-driven allocation strategy (Eq. 11) in E-step.

## 3.5 Entity-level Information Injection via MSSG

To fully exploit fine-grained inter-modality features at the entity level, we propose the construction of a multimodal-semantic scene graph. This MSSG utilizes balanced multimodal features to enable entity-level fine-grained interactions, as illustrated in Figures 2 and 3. Our MSSG is an extension of the vanilla semantic scene graph

that represents the intrinsic relationships between entities under textual semantics. To incorporate collaborative modality effects into representation learning, we concatenate additional modal features (such as video and audio features) into entities within the traditional scene graph to obtain a structured multimodal context. Specifically, we employ a scene graph parser [46] [1] to initially generate the textual scene graphs $\mathcal{G}^{\mathbf{T}} = \{\mathcal{E}^{\mathbf{T}}, \mathcal{R}^{\mathbf{T}}\}$, where $\mathcal{E}^{\mathbf{T}}$ represents entities and $\mathcal{R}^{\mathbf{T}}$ denotes relation within textual descriptions. Subsequently, we augment these textual entities $\mathcal{E}^{\mathbf{T}}$ by aggregating multimodal-entity features to construct the MSSG $\mathcal{G}^{\mathbf{M}} = \{\mathcal{E}^{\mathbf{M}}, \mathcal{R}^{\mathbf{M}}\}$, where,

- $\mathcal{E}^{\mathbf{M}} = \{[\overset{\mathcal{M}\backslash\{\mathbf{T}\}}{\underset{m}{||}} h_i^m, e_i^{\mathbf{T}}]\}, \forall e_i^{\mathbf{T}} \in \mathcal{E}^{\mathbf{T}}$, represents the entity set aggregated with multimodal semantics, where, $||$ is a concatenation operation.

- $\mathcal{R}^{\mathbf{M}} = \mathcal{R}^{\mathbf{T}}$ denotes the set of the edges connecting nodes in the scene graph.

MSSG provides key insights for achieving fine-grained, entity-level multimodal interactions. However, the heterogeneous nature of MSSG, featuring diverse node types (such as 'dogs' and 'clothes') and relational edges (such as 'above' and 'below'), presents challenges for traditional graph processing techniques, such as graph convolution networks (GCN) [18, 51] or graph attention networks (GAT) [41]. These standard approaches often struggle to capture the diversity of nodes and edges in heterogeneous graphs [35] To achieve fine-grained aggregation in our heterogeneous MSSG $\mathcal{G}^{\mathbf{M}}$, we propose utilizing heterogeneous graph attention networks (HGAT) to distinguish specific information from different types of nodes along different relation edges. Specifically, we feed node features $H_i \in \mathcal{E}^{\mathbf{M}}$ into HGAT with $K$ attention heads to obtain attention scores of edge $r$ on the $k$-th head as:

$$\mathcal{F}\left(H_i, H_j\right) = \mathbf{a}^{[k,r]\top}\left[W^{[k,r]}H_i \| W^{[k,r]}H_j\right], \quad (12)$$

$$\alpha_{ij}^{[k,r]} = \frac{\exp\left(\sigma\left(\mathcal{F}\left(H_i, H_j\right)\right)\right)}{\sum_{j' \in \mathcal{N}_i} \exp\left(\sigma\left(\mathcal{F}\left(H_i, H_{j'}\right)\right)\right)}, \quad (13)$$

where, $||$ is a concatenation operation, $r$ denotes the type of edge from node $j$ to node $i$; $\sigma$ represents the activation function; the set $\mathcal{N}_i$ is the first-order neighbors of node $i$ on graph $\mathcal{G}^{\mathbf{M}}$; and $\mathbf{a}^{[k,r]}$ and $W^{[k,r]}$ are trainable matrix of $r$ on the $k$-th head. The core difference between HGAT in Eq. 12 and GAT is that we assign a learnable weight $W^{[k,r]}$ to each edge, rather than sharing a set of parameters. We can further obtain the enhanced multimodal representation as follows:

$$\hat{H}_i = \overset{K}{\underset{k=1}{||}} \sigma\left(\sum_{j \in \mathcal{N}_i} \alpha_{ij}^{[k,r]} W^{[k,r]} H_j\right). \quad (14)$$

Then we decouple the enhanced features $\hat{H} = \{\hat{H}_i\}$ into modality-specific representations $h^{\mathbf{T}}, h^{\mathbf{I}}, h^{\mathbf{A}}$ and $h^{\mathbf{V}}$ for M-step in Section 3.3.

We summarize the whole training process of the EM framework in Algorithm 1. It is worth noting that our method proposes a framework wherein probability estimation can be flexibly adapted to suit a wide range of scenarios through various approaches.

---

**Algorithm 1** Training algorithm for MASE

**Input:** The dataset $\mathcal{D} = \{x_i, y_i\}_{i=1}^n$ for training.
**Parameter:** Learnable model parameters $\Theta$.
1: Initialize $\{P_i\}_{i=1}^n$ to uniform distribution.
2: **for** each batch $\mathcal{B}$ in $\mathcal{D}$ **do**
3:     Compute modal semantics $\{h^m\}$;
4:     **while** not converged **do**
5:         Estimate $\{P_i\}_{i=1}^n$ via Eq. 11; \\ E-step
6:         Obtain $\hat{h}^m, \forall m \in \mathcal{M}$ via Eq. 14; \\ E-step
7:         Update $\Theta$ via Eq. 9. \\ M-step
8:     **end while**
9: **end for**
10: **return** The optimal parameters $\Theta^*$.

---

## 4 EXPERIMENTS

### 4.1 Experimental Setup

**Datasets.** We conduct main experiments across four types of tasks: **(1) Natural Language Understanding (NLU).** We evaluate our MASE on six GLUE benchmark [42], i.e., SST-2 [34], QNLI [32], QQP[6] , MNLI [44], MRPC [10], and STS-B [5]. **(2) Question Answering.** In the realm of question answer task, we select CSQA [37] and SQuADv2.0 [31] to evaluate MASE. **(3) Text Generation.** We utilize the CommonGen [22] as the dataset to validate MASE. **(4) Visual Reasoning.** We evaluate our MASE on color reasoning (using the MemoryColor [27] and ColorTerm [4] datasets) and shape reasoning (using ObjectShape [50] dataset) tasks. **(5) Visual Question Answering.** We further evaluate our MASE on cross-modal question-answering tasks on VQA 2.0 benchmark [13].

**Baselines.** We compare MASE with the pre-trained language models (PLMs), the multimodal proxy based baselines (MPB), multimodal contrastive pre-trained models (MC-PTMs), and visually-augmented pre-trained language models (VA-PLMs). (1) PLMs: We utilize BERT [9], RoBERTa [23], XLNet [49], and T5 [30] as the backbones, and directly fine-tune them as baselines. (2) MPB: We use the methods described in Section 3.1 as the baseline of multimodal proxy based MA-PLMs. (3) MC-PTMs: CLIP [29], AudioCLIP [16] and CLIP-ViP [47] are selected as baselines. (4) VA-PLMs: We select VOKEN [38], iACE [25], and VAWI [14] as baselines.

**Implementation Details.** We implement all methods[2] based on Huggingface Transformers [45]. The number of iterations for the EM algorithm is set to 10. We set the learning rate as 1e-4 on GLUE benchmark, 3e-5 on CSQA and SQuADv2.0 datasets, and 2e-5 on CommonGen. We utilize Adam as the optimizer and train all models for 3 epochs. $\{P_i\}_{i=1}^n$ is initialized as a uniform distribution, i.e., the initial weight of each modality is 0.25 for four modalities. The experimental details on VQA2.0 are consistent with ConceptBert [11]. **More details and experiments can be found in supplements**.

### 4.2 Main Results

**Evaluation on NLU Tasks.** Experimental results on GLUE are shown in Table 1, from which we have several observations:

(1) The integration of additional multimodal information significantly bolsters the predictive capabilities of PLMs in NLU tasks.

---

[1] https://github.com/vacancy/SceneGraphParser

[2] All the source code and models will be released after review.

**Table 1: Comparison of accuracy and learnable parameter size on the GLUE benchmark, with the best results highlighted in bold. "T, I, A, V" indicates four distinct modalities: text, image, audio, and video, respectively. We gradually add different modal semantics to MASE. "+None" denotes we directly fine-tune the base model. The results of VOKEN, iACE and VAWI on GLUE are reported from Lu et al. [25] and Guo et al. [14]. More results on other base models are provided in the supplements.**

| Base Model | Method | Modality | SST-2 | QNLI | QQP | MNLI | MRPC | STS-B | Average | Param. |
|---|---|---|---|---|---|---|---|---|---|---|
| BERT-base [9] | +None | T | 89.3 | 87.9 | 87.2 | 79.4 | 81.7 | 84.4 | 84.98 | 110M |
| | +VOKEN [38] | T+I | 92.2 | 88.6 | 88.6 | 82.6 | 83.5 | 86.0 | 86.83 | 121M |
| | +iACE [25] | T+I | 91.7 | 88.6 | 89.1 | 82.8 | 85.8 | 86.6 | 87.43 | 568M |
| | +VAWI [14] | T+I | 92.4 | 89.1 | 89.7 | 83.0 | 85.6 | 86.9 | 87.78 | 156M |
| | +MPB | T+I+A+V | 91.0 | 89.1 | 88.5 | 82.3 | 83.4 | 85.7 | 86.67 | 118M |
| | +MASE | T+I | 93.1 | 89.7 | 90.7 | 83.4 | 87.0 | 87.6 | 88.58 | 135M |
| | +MASE | T+I+A | 93.4 | **91.5** | 91.1 | 84.0 | 87.2 | 87.7 | 89.15 | 135M |
| | +MASE | T+I+A+V | **93.9** | 91.4 | **91.6** | **84.5** | **87.5** | **88.4** | **89.55** | 135M |
| RoBERTa-base [23] | +None | T | 89.2 | 87.5 | 86.2 | 79.0 | 81.4 | 85.4 | 84.78 | 355M |
| | +VOKEN [38] | T+I | 90.5 | 89.2 | 87.8 | 81.0 | 87.0 | 86.9 | 87.06 | 367M |
| | +iACE [25] | T+I | 91.6 | 89.1 | 87.9 | 82.6 | 87.7 | 86.9 | 87.06 | 738M |
| | +VAWI [14] | T+I | 91.6 | 90.6 | 87.9 | 82.4 | 88.5 | 88.3 | 88.21 | 402M |
| | +MPB | T+I+A+V | 90.8 | 89.7 | 88.1 | 81.4 | 85.6 | 86.7 | 87.05 | 363M |
| | +MASE | T+I | 93.1 | 91.5 | 88.6 | 84.0 | 89.1 | 89.0 | 89.22 | 383M |
| | +MASE | T+I+A | 93.2 | 91.8 | 89.2 | 84.7 | 89.5 | **89.9** | 89.72 | 383M |
| | +MASE | T+I+A+V | **93.7** | **92.6** | **89.8** | **85.1** | **90.3** | 89.5 | **90.12** | 383M |

**Table 2: Performance comparison on CommonGen, with the best results highlighted in bold. We use T5-3b as the backbone. "+Image" denotes we utilize retrieval-based methods through Bing web API similar to Table 1.**

| Method | BLUE-4 | METOR | Rouge-L | CIDER |
|---|---|---|---|---|
| T5-3b [30] | 36.2 | 32.7 | 59.3 | 17.7 |
| +Image | 35.8 | 32.1 | 59.0 | 17.6 |
| +MASE | **38.3** | **34.1** | **62.4** | **18.9** |

MPB, VA-PLMs and our MASE achieve substantial performance gains consistently across all backbone networks. This indicates that introducing general object knowledge (e.g., color and shape) through additional modalities can markedly improve PLMs.

(2) MASE demonstrates superior performance compared to MPB and all VA-PLM baselines when the same number of additional modalities are incorporated. We attribute this to (i) the efficient injection of semantics and the balance of modal information through our MASE framework; (ii) the strategy of incorporating multimodal information extracted through multimodal proxy based methods to avoid the inclusion of noise and modality gaps.

(3) Furthermore, MASE surpasses all baselines in terms of PLM performance gains across all backbone networks, which demonstrates the effectiveness and generality of our MASE.

**Evaluation on Text Generation Tasks.** We conduct experiments on the CommonGen dataset as results shown in Table 2. We use T5-3B as the backbone network and replace the optimization objective of M-step with the text generation objective. We can observe that MASE performs better in all metrics than the base model and retrieved-based methods. This suggests that our method can aggregate multimodal information into PLMs for better text generation.

**Evaluation on Question Answering Tasks.** We present our experimental results on CSQA and SQuAD v2 datasets in Table 3. For comparison with the retrieval-based methods, we adopt Bing Image Search[3] for image retrieval and utilize the CLIP image encoder to extract visual features. We also evaluate different approaches in low-resource settings. Analysis of Table 3 yields several insights:

(1) MASE significantly improves performance at low-resource settings (i.e., with only 5% of data available). This demonstrates that MASE can integrate balanced additional information into PLM, effectively countering the adverse effects of data scarcity.

(2) We can observe that the performance gain obtained by retrieval-based methods is lower than MASE. Employing our multimodal enhancement proves to be more efficient and low-cost compared to retrieval-based strategies.

(3) In the realm of QA, MASE can achieve significant performance gains across a variety of datasets and multiple foundational models, including BERT, RoBERTa, and XLNet. This further validates that MASE offers a twofold advantage: it can introduce additional modal information in a balanced and efficient manner and it is compatible with different architectures and various scenarios.

**Evaluation on Visual Reasoning Tasks.** We utilize color and shape reasoning datasets, i.e., the MemoryColor [27], ColorTerm [4], and ObjectShape [50] datasets, for evaluating visual knowledge transfer of our MASE, as results shown in Table 4. Based on these results, we observe that MASE positively contributes to the understanding of object colors and shapes for PLMs, demonstrating the effectiveness of our approach in enhancing the multimodal comprehension abilities of PLMs.

**Evaluation on Visual Question Answering Tasks.** To verify the effectiveness of our MASE on cross-modal QA, we conduct

---

[3]https://learn.microsoft.com/en-us/azure/cognitive-services/bing-image-search/overview

**Table 3: Comparison of various methods on CSQA and SQuAD v2.0 datasets, with the best results highlighted in bold. We randomly select 5% of the samples from datasets to evaluate different methods. "+Image" denotes we introduce retrieved images through Bing web search API and utilize CLIP as image feature extractors.**

| Base Model | Method | Modality | CSQA | | | | SQuAD v2 | | | |
|---|---|---|---|---|---|---|---|---|---|---|
| | | | 5% | | 100% | | 5% | | 100% | |
| | | | Acc. | F1-score | Acc. | F1-score | Acc. | F1-score | Acc. | F1-score |
| BERT-base [9] | +None | T | 65.6 | 50.3 | 81.6 | 68.6 | 54.8 | 57.9 | 72.1 | 75.2 |
| | +Image (iACE) | T+I | 66.7 | 51.2 | 82.3 | 69.9 | 56.5 | 59.1 | 73.4 | 76.0 |
| | +MPB | T+I+A+V | 67.6 | 51.8 | 82.3 | 69.6 | 57.4 | 59.6 | 74.8 | 77.5 |
| | +MASE | T+I | 68.5 | 53.6 | 83.7 | 71.5 | 58.8 | 63.2 | 75.6 | 79.1 |
| | +MASE | T+I+A+V | **70.9** | **55.2** | **86.8** | **73.3** | **61.7** | **65.3** | **78.4** | **82.3** |
| RoBERTa-base [23] | +None | T | 70.9 | 56.6 | 83.3 | 72.6 | 62.6 | 68.5 | 77.6 | 81.2 |
| | +Image (iACE) | T+I | 71.3 | 57.0 | 84.0 | 73.6 | 63.3 | 69.1 | 78.0 | 81.7 |
| | +MPB | T+I+A+V | 72.1 | 57.5 | 84.3 | 74.0 | 63.1 | 68.7 | 79.7 | 82.9 |
| | +MASE | T+I | 73.4 | 58.6 | 86.2 | 76.1 | 65.3 | 71.5 | 80.6 | 83.9 |
| | +MASE | T+I+A+V | **76.2** | **59.8** | **87.7** | **77.8** | **68.6** | **73.3** | **82.8** | **85.7** |
| XLNet-large [49] | +None | T | 75.2 | 61.0 | 86.4 | 75.6 | 68.8 | 72.0 | 79.4 | 82.6 |
| | +Image (iACE) | T+I | 76.2 | 61.9 | 87.1 | 76.3 | 69.2 | 73.7 | 79.8 | 82.9 |
| | +MPB | T+I+A+V | 76.5 | 61.9 | 88.0 | 76.8 | 70.0 | 74.1 | 80.2 | 83.3 |
| | +MASE | T+I | 77.7 | 63.0 | 88.8 | 77.4 | 72.1 | 76.1 | 81.5 | 84.4 |
| | +MASE | T+I+A+V | **79.4** | **64.6** | **89.2** | **78.7** | **73.6** | **77.3** | **83.0** | **86.4** |

**Table 4: Comparison of various methods on MemoryColor, ColorTerm, and ObjectShape datasets, with the best results highlighted in bold.**

| Method | MemoryColor | ColorTerm | ObjectShape |
|---|---|---|---|
| CLIP | 27.3 | 24.9 | 19.8 |
| BERT-base | 25.1 | 26.7 | 31.5 |
| RoBERTa-base | 26.9 | 25.4 | 32.3 |
| BERT-base + MASE | **37.1** | **35.8** | **35.7** |

**Table 5: Comparison of various methods on VQA v2.0 validation set. '+MASE' denotes the addition of our MASE to the PLMs (i.e. BERT) to introduce multimodal semantics.**

| Model | Method | Overall | Yes/No | Number | Other |
|---|---|---|---|---|---|
| Up-Down [1] | +None | 59.6 | 80.3 | 42.8 | 55.8 |
| XNM Net [33] | +None | 64.7 | - | - | - |
| ReGAT [20] | +None | 67.2 | - | - | - |
| ViLBERT [24] | +None | 67.9 | 82.6 | 54.3 | 67.2 |
| SIMPLE [11] | +None | 67.9 | 82.7 | 54.4 | 67.2 |
| CONCAT [11] | +None | 68.1 | 83.0 | 54.6 | 68.0 |
| ConceptBert [11] | +None | 70.0 | 84.0 | 55.3 | 70.6 |
| ViLBERT | **+MASE** | 68.5 | 84.0 | 54.9 | 70.4 |
| ConceptBert | **+MASE** | 72.3 | 84.9 | 56.5 | 73.1 |

experiments on the VQA 2.0 benchmark and compare with baselines, as shown in Figure 5. We can observe significant VQA performance gains brought about by our MASE, e.g., +1.6% and +2.3% overall accuracy using ViLBERT and ConceptBert as base methods. This validates that our MASE can inject multimodal semantics into PLMs and improve their multimodal understanding ability, which

significantly improves the performance in the cross-modal visual question-answering task.

## 4.3 Ablation Study

**The Effect of the Cross-modal Contrastive Loss in M-step.** We conduct experiments to study the cross-modal contrastive loss of MASE in M-step as illustrated in Table 6. It can be seen that the performance of MASE significantly decreases without the cross-modal contrastive loss to improve the information aggregation. This verifies that (1) the optimization algorithm of MASE in M-step is capable of injecting additional modality information sufficiently and effectively. (2) the interaction of different modalities plays a significant role in enhancing the expressive capabilities of PLMs.

**The Effect of the Information-driven Self-balancing in E-step.** We employ BERT-base and RoBERTa-base as the backbones and conduct experiments to study the impact of our information-driven self-balancing strategy in E-step. Observations indicate that the absence of our information-driven self-balancing strategy significantly diminishes performance. This suggests that our information-driven self-balancing strategy can dynamically estimate the contribution of different modalities and is effective and balanced in transferring multimodal knowledge to PLMs.

**Combined with Prompt Learning.** We combined MASE with prompt learning [36] to evaluate the effectiveness of MASE under the parameter-efficient fine-tuning approach. As shown in Table 6, we can observe significant performance gains from MASE compared to the MPB baseline using the prompt learning strategy. This shows that MASE can be effectively compatible with existing efficient fine-tuning methods to improve the performance of PLMs. This experiment further demonstrates the great potential of MASE in practical scenarios.

**Dynamic Analysis of Latent Variables** To study the effect of our EM-based iteration, we visualize the changes in the latent variable

**Table 6: Ablation experiments on the GLUE benchmark. "+Prompt Learning" denotes we use prompt learning to achieve efficient parameter fine-tuning for PLMs. "+None" denotes utilizing the text encoder of MC-PTMs for prediction.**

| Base Model | Method | SST-2 | QNLI | QQP | MNLI | MRPC | STS-B | Average |
|---|---|---|---|---|---|---|---|---|
| CLIP | +None | 72.4 | 72.6 | 70.2 | 69.1 | 73.9 | 75.1 | 72.22 |
| BERT-base [9] | MPB + Prompt Learning | 86.4 | 87.7 | 86.0 | 76.9 | 64.9 | 84.4 | 81.05 |
| | MASE + Prompt Learning | 89.2 | 89.4 | 88.6 | 79.1 | 68.2 | 86.0 | 83.42 |
| | w/o Contrastive Loss in M-step | 92.9 | 90.7 | 90.7 | 83.4 | 86.6 | 87.5 | 88.63 |
| | w/o Self-balancing in E-step | 92.6 | 89.0 | 90.3 | 82.7 | 85.8 | 86.8 | 87.87 |
| | w/o Entity-level Injection in E-step | 93.0 | 90.2 | 90.3 | 83.1 | 86.2 | 87.3 | 88.35 |
| | Full MASE | **93.9** | **91.4** | **91.6** | **84.5** | **87.5** | **88.4** | **89.55** |
| RoBERTa-base [23] | MPB + Prompt Learning | 85.5 | 88.8 | 85.7 | 77.5 | 67.2 | 84.9 | 81.60 |
| | MASE + Prompt Learning | 88.7 | 90.3 | 87.6 | 80.7 | 70.3 | 86.2 | 83.97 |
| | w/o Contrastive Loss in M-step | 92.9 | 91.4 | 88.8 | 84.0 | 89.0 | 87.9 | 89.00 |
| | w/o Self-balancing in E-step | 92.6 | 91.6 | 88.3 | 83.5 | 88.5 | 88.1 | 88.77 |
| | w/o Entity-level Injection in E-step | 93.0 | 91.3 | 88.0 | 83.8 | 88.7 | 88.4 | 88.87 |
| | Full MASE | **93.7** | **92.6** | **89.8** | **85.1** | **90.3** | **89.5** | **90.12** |

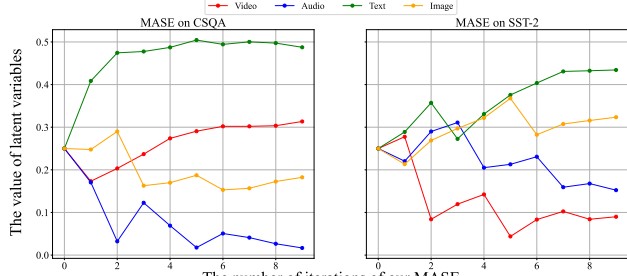

**Figure 4: We visually demonstrate the changes in modal information allocation weights during the optimization process of our MASE on the CSQA (left) and SST-2 (right) dataset.**

during each E-step of the training process on CSQA and SQuAD v2 datasets. We use RoBERT-base as the base model. Specifically, we report the latent variables averaged over four different modalities as shown in Figure 4, and observe several important phenomena:

(1) The contribution of each modality to the task varies. We can observe that the weights of different modalities are different on CSQA and SST-2 datasets. This indicates that optimizing the balance of modal information is crucial for improving the multimodal understanding ability of PLMs.

(2) The weight of information allocation will undergo dynamic changes during the iteration stage. In the early stage of EM iteration, there is a significant change in the allocation of weights, which is due to insufficient optimization of model parameters in the early stage. After the model parameters are fully optimized, the rate of change of the latent variables becomes gentle, which proves that the modal allocation has reached equilibrium.

**Feature Visualization** We perform T-SNE visualization on the classification and specific-modal features of different methods at the representation level, and observe that: (1) Our MASE can achieve the improvement of intra-class consistency and inter-class discrimination of representations by introducing multimodal knowledge,

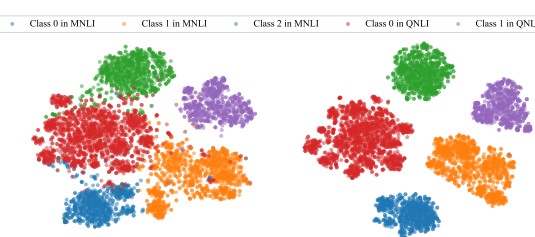

**(a) Visualization of MPB (left) and MASE (right) on MNLI and QNLI.**

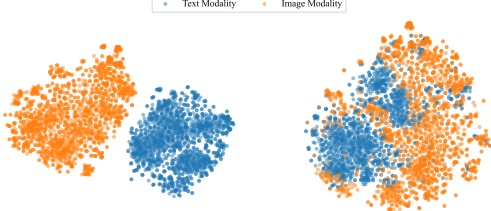

**(b) Visualization of real images (left) and our proxies (right) on CSQA.**

**Figure 5: T-SNE visualization [40] of feature discriminative (a) and modality gaps (b) with RoBERTa as the base model.**

as shown in Figure 5a. (2) Our method can effectively alleviate modality gaps, such as images and texts, as shown in Figure 5b.

## 5 CONCLUSION

In this paper, we present a novel principled self-balancing probabilistic framework, MASE, designed for injecting multimodal semantics into PLMs. MASE assigns weights to modal information as latent variables and adopts an EM-based iterative algorithm to iteratively optimize the two objectives of multimodal information injection and balance. Furthermore, we propose a novel MI-driven allocation estimation method and achieve entity-level multimodal interactions via MSSG. The experimental results show that MASE can effectively improve the performance of PLMs on multiple tasks, and is compatible with efficient fine-tuning strategies.

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
