# OpenReview forum: "Towards Multimodal-augmented Pre-trained Language Models via Self-balanced Expectation-Maximization Iteration"
_acmmm.org/ACMMM/2024/Conference — MM2024 Poster_

### Official Review · Reviewer_zjcx · 2024-05-10

**Rating:** 4
**Confidence:** 2

**Summary:**

This paper introduces a principled self-balancing probabilistic framework called MASE, designed to infuse multimodal semantics into PLMs. Utilizing latent variables, MASE assigns weights to modal information and employs an EM-based iterative algorithm to optimize the two objectives of multimodal information infusion and balancing. Additionally, this paper suggests an MI-driven allocation estimation method to accomplish entity-level multimodal interactions through MSSG. Ultimately, the authors conduct numerous experiments, thereby validating the effectiveness of the proposed framework.

**Strengths:**

1.The MASE framework proposed in this paper is novel.
2.The authors conduct extensive experiments to demonstrate that MASE can effectively improve the performance of PLMs on multiple tasks.
3.The logic of this paper is relatively smooth.

**Limitations:**

1. The authors concatenate multimodal features with entity nodes in the vanilla scene graph to construct MSSG. However, in the ablation study, experiments involving the removal of entity nodes are lacking.
2. It is suggested that the font format in Figure 3 be unified.

**Suitability:**

3

---

### Official Review · Reviewer_uQua · 2024-05-21

**Rating:** 4
**Confidence:** 2

**Summary:**

This paper presents MASE, an innovative iterative framework for multimodal-augmented pre-trained language models (PLMs) to overcome the limitations of existing methods that face modality gaps, noise, and neglect of key visual or acoustic semantics. Experimental results show MASE significantly outperforms baselines across various tasks and architectures and is compatible with efficient parameter fine-tuning methods like prompt learning.

**Strengths:**

- MASE introduces a novel iterative framework utilizing an EM-based algorithm to balance and efficiently inject multimodal semantics into PLMs, which is the first theoretical framework specifically designed for multimodal-augmented PLMs.
- The paper proposes a mutual information-based information allocation estimation method and an entity-level multimodal aggregation method via the multimodal-semantic scene graph (MSSG) to enhance modal semantics. The method is introduced in great detail, providing clear explanations of each component and process within the MASE framework.
- Extensive experiments on five tasks, including NLP, visual reasoning, and visual question answering, demonstrate MASE's effectiveness and generality. The paper includes a wide range of experiments, thoroughly validating the proposed method's performance and robustness across multiple tasks and scenarios.

**Limitations:**

- The related work section of the paper is relatively brief, potentially lacking in comprehensive coverage of existing methods and approaches in the field.
- The paper does not explicitly address the performance of the proposed method, MASE, on widely used settings such as BERT-large and RoBERTa-large. This omission leaves a gap in understanding how well MASE scales and adapts to these larger, more commonly used models.
- Typo:  "balance and efficiently" in 177 line, "Figure 5" -> "Table 5" in 749 line, "+1.6%" -> "+0.6%" in 749 line

**Suitability:**

2

---

### Official Review · Reviewer_CzTF · 2024-05-25

**Rating:** 5
**Confidence:** 3

**Summary:**

This paper presents MASE, a novel framework for enhancing pre-trained language models (PLMs) with multimodal information. MASE addresses issues like modality gaps, noise, and entity-level semantics by using an Expectation Maximization (EM) algorithm and multimodal proxies. It integrates a mutual information-based allocation strategy and constructs a multimodal-semantic scene graph (MSSG) for better entity-level understanding. Experiments show MASE's effectiveness across multiple tasks and its compatibility with existing fine-tuning methods.

**Strengths:**

1. The introduction of MASE presents a significant advancement in integrating multimodal information into PLMs, addressing key issues like modality gaps and noise with a principled EM-based iterative framework.

2. The use of multimodal proxies, mutual information-based allocation estimation, and multimodal-semantic scene graphs are innovative methods that enhance the understanding and performance of PLMs.

3. Extensive experiments on thirteen datasets across five tasks validate the effectiveness and versatility of MASE, showing its consistent outperformance against competitive baselines.

**Limitations:**

1. Could the implementation of the EM-based iterative framework and multimodal-semantic scene graphs be computationally intensive? How does this affect the scalability of the method to very large datasets or real-time applications?

2. While the framework aims for balanced multimodal integration, could certain modalities benefit PLMs more significantly than others? Have the authors analyzed the impact of individual modalities on performance? What are the results for T+A and T+V combinations in Tables 1 and 3?

**Suitability:**

3

---

### Meta-Review · Area_Chair_Dj5Z · 2024-07-02

**Recommendation:** Accept (Poster)
**Confidence:** 5

**Metareview:**

This work proposes the MASE framework, an iterative-based approach for multimodal-augmented pre-trained language models, leveraging Expectation-Maximization to balance the introduction of multimodal semantics, in a principled manner.

Overall, after the rebuttal, the paper got three positive reviews, having received two Weak Accepts and one Borderline Accept.

Reviewers unanimously agree on the novelty of the MASE framework and on the extensive experimental evaluation carried out by the authors.

Among its weaknesses, reviewers mentioned the brief related work section (uQua), followed by some ablation studies that are missing. Namely, whether certain modalities could have a bigger benefit than others (CzTF), and variants of the MSSG graph (zjcx).

In summary, while the paper could be improved with further ablation studies, reviewers recognize that the conducted experiments validate the proposed approach. Moreover, reviewers recognize the principled nature of the proposed Expectation-Maximization approach, in jointly addressing inter-modality gaps, noise, and entity-level semantics. Therefore, I suggest this work to be accepted as Poster.